# Emerging Breast Cancer Subpopulations: Functional Heterogeneity Beyond the Classical Subtypes

**DOI:** 10.3390/ijms262311599

**Published:** 2025-11-29

**Authors:** Amalia Kotsifaki, Georgia Kalouda, Efthymios Karalexis, Martha Stathaki, Georgios Metaxas, Athanasios Armakolas

**Affiliations:** 1Physiology Laboratory, Medical School, National and Kapodistrian University of Athens, 11527 Athens, Greece; amkotsifaki@med.uoa.gr (A.K.); gkalouda@med.uoa.gr (G.K.); euthimiskaralexis12@gmail.com (E.K.); 2Department of Surgery, Elena Venizelou Hospital, 11521 Athens, Greece; stathakimg@yahoo.gr (M.S.); geometa@hotmail.com (G.M.)

**Keywords:** breast cancer subtypes, emerging populations, functional heterogeneity, HER2-low, claudin-low, BRCAness, prognostic biomarkers, tumor microenvironment, precision oncology, intratumoral heterogeneity

## Abstract

Breast cancer (BC) is increasingly recognized as a heterogeneous disease, with complexity that extends beyond the classical luminal A/B, HER2-enriched, and triple-negative framework. Advances in molecular and functional profiling have uncovered emerging subpopulations, including HER2-low, claudin-low, BRCA-deficient (“BRCAness”), and refined TNBC subsets, such as luminal AR (LAR) and basal-like immune variants, that extend beyond traditional taxonomies. These novel classifications provide additional resolutions, offering both prognostic insight and therapeutic opportunities. In this comprehensive review, we integrate evidence from genomic, epigenetic, proteomic, immune-related, and liquid biopsy biomarkers, underscoring how they define the biology of these subgroups and predict responses to targeted therapies, such as antibody–drug conjugates, PARP inhibitors, and immune checkpoint blockade. We further highlight the role of the tumor microenvironment (TME) and intratumoral heterogeneity in shaping these entities. Collectively, recognition of emerging subtypes as clinically actionable groups represents a paradigm shift from static receptor-based models to dynamic, biomarker-driven frameworks that refine prognosis, enable patient stratification, and support precision oncology in aggressive BC.

## 1. Introduction

Breast cancer (BC) remains the most frequently diagnosed malignancy in women and one of the leading causes of cancer-related mortality worldwide [1,2]. According to the World Health Organization (WHO), approximately 7.8 million women were diagnosed within the last five years, with 2.3 million new cases and nearly 685,000 deaths recorded in 2020 alone [3]. Beyond incidence and mortality, the disease imposes a substantial overall health burden, exceeding 20 million disability-adjusted life years (DALYs) worldwide, an increase from the 15.1 million DALYs recorded in 2019, highlighting its growing impact on both mortality and long-term morbidity [4]. These DALYs represent both premature loss of life and the accumulation of years lived with treatment-related or disease-related morbidity, including chronic complications, functional limitations, and long-term effects of systemic therapy [5].

Although far less common, male BC (MBC) warrants attention. Representing roughly 1% of all BC cases, MBC exhibits distinct biological and clinical characteristics, including a predominance of hormone receptor–positive tumors and specific genetic vulnerabilities, such as BRCA2 mutations [6,7]. Integrating up-to-date insights on MBC enhances our understanding of BC heterogeneity, helps address an often-underrepresented patient group and highlights the urgent need for more effective strategies in classification, prognosis, and treatment [8,9].

Historically, BC classification was grounded in morphological criteria, differentiating non-invasive forms, such as ductal carcinoma in situ (DCIS) and lobular carcinoma in situ (LCIS) from invasive types, most notably invasive ductal carcinoma (IDC) and invasive lobular carcinoma (ILC) [10]. While histopathology laid the foundation for BC taxonomy, it soon became evident that morphology alone could not account for the marked clinical variability among patients, prompting the shift toward molecular profiling [10].

The paradigm shifted with the advent of molecular profiling, which enabled stratification of BCs into biologically meaningful groups defined by the expression of estrogen receptor (ER), progesterone receptor (PR), and human epidermal growth factor receptor 2 (HER2) [2]. These biomarkers, readily detected by immunohistochemistry, established the foundation for four broadly recognized clinical subtypes: luminal A, luminal B, HER2-enriched, and Triple-negative BC (TNBC) [11,12]. Luminal A tumors, typically ER- and PR-positive with low Ki-67, display indolent behavior and favorable prognosis, accounting for nearly 50–60% of cases [10,13]. Luminal B tumors, though also hormone receptor-positive, exhibit higher proliferative indices and more aggressive features, with poorer outcomes relative to luminal A [14]. HER2-positive tumors, defined by ERBB2 amplification or overexpression, historically portended a dismal prognosis until the advent of HER2-targeted therapies, most notably trastuzumab dramatically improved outcomes [3]. TNBC, lacking ER, PR, and HER2 expression, remains the most aggressive and therapeutically challenging subtype, with high rates of recurrence and metastasis, particularly in younger women [2,11,15].

Although these intrinsic categories have revolutionized BC management by guiding endocrine, targeted, and chemotherapeutic strategies, they fail to capture the full spectrum of heterogeneity. Both intertumor and intratumor heterogeneity significantly influence disease progression, therapeutic resistance, and survival outcomes [10,16]. Intertumor heterogeneity arises from genomic, transcriptomic, and phenotypic differences across patients, whereas intratumor heterogeneity, driven by clonal diversity, epigenetic plasticity, and interactions with the tumor microenvironment (ΤΜΕ), accounts for the variable clinical behavior observed among tumors classified within the same subtype [17]. More recently, the concept of functional heterogeneity has emerged, capturing the observation that even tumors with highly similar molecular profiles may diverge in biological behavior, treatment response, and metastatic potential [18]. Collectively, these observations expose the limitations of the classical four-subtype model [17].

Among the most compelling developments is the recognition of HER2-low BC as a biologically distinct entity. Traditionally grouped within the HER2-negative category, HER2-low tumors are characterized by immunohistochemistry scores of 1+ or 2+ without ERBB2 amplification [19,20]. Recent clinical trials with novel antibody–drug conjugates have demonstrated therapeutic benefit in this subgroup, highlighting its clinical relevance and challenging the binary HER2-positive/negative paradigm [21]. Similarly, claudin-low tumors, enriched in features of epithelial-to-mesenchymal transition (EMT), stemness, and immune infiltration, represent an additional subtype with poor prognosis and unique biological behavior [22,23]. These findings illustrate that BC is inadequately captured by the four canonical categories [21].

The complexity of TNBC underscores the necessity for more refined subtyping [23]. Once regarded as a single disease entity, TNBC is now understood to encompass a spectrum of biologically diverse tumors, each characterized by distinct molecular signatures and clinical trajectories [24]. Early gene expression profiling studies proposed six to seven TNBC subtypes, including basal-like 1 (BL1), basal-like 2 (BL2), immunomodulatory (IM), mesenchymal (M), mesenchymal stem-like (MSL), luminal androgen receptor (LAR), and an unstable subtype [15,22,25]. These initial classifications have since been consolidated into fewer but more clinically meaningful subgroups: basal-like immune-activated (BLIA), basal-like immunosuppressed (BLIS), mesenchymal (MES), and LAR [26,27,28]. Each of these subtypes exhibits distinct biological features and therapeutic susceptibility [29]. For example, BL1 tumors, marked by robust DNA damage response activity, tend to be sensitive to platinum-based chemotherapy, whereas immunomodulatory tumors, enriched in tumor-infiltrating lymphocytes (TILs) and characterized by immune checkpoint activation, are more likely to respond to immunotherapy [24]. In contrast, mesenchymal and mesenchymal stem-like tumors display EMT and stemness-associated traits, which are linked to poor prognosis and resistance to conventional therapies [24,29,30].

Emerging multi-omics and proteogenomic approaches have further deepened insights into TNBC heterogeneity [31]. Integrated analyses combining genomics, transcriptomics, proteomics, and metabolomics have identified novel biomarkers predictive of treatment response and resistance, such as proteomic signatures linked to chemotherapy resistance and residual disease [16,29]. Single-cell RNA sequencing and spatial transcriptomics have uncovered subpopulations of basal epithelial cells enriched for EMT programs and chemoresistance-associated pathways, underscoring the intratumoral complexity of TNBC [30]. These approaches not only enhance subtype classification but also provide actionable biomarkers for stratifying patients into targeted therapies [32].

The role of the TME is another dimension of heterogeneity that has gained prominence [33]. TNBC tumors, for instance, may be stratified into inflamed, immune-excluded, or immune-desert phenotypes based on TIL distribution, each correlating with distinct prognoses and responses to immunotherapies [27]. Integrative classifications that combine intrinsic molecular features with TME signatures are increasingly viewed as essential for guiding effective treatment decisions [34]. Furthermore, circulating biomarkers such as circulating tumor cells (CTCs), cell-free DNA (cfDNA), and non-coding RNAs [microRNAs (miRNAs), lncRNAs] emerging as minimally invasive tools for detecting subtypes, monitoring therapeutic responses, and predicting recurrence [3]. The incorporation of such liquid biopsy–based biomarkers could significantly enhance the precision of subtype classification and patient management [1,2].

Importantly, these developments have practical implications for therapy. HER2-low classification has already altered clinical trial design and regulatory approvals, while proteogenomic stratification promises to refine drug selection within TNBC [11,21]. Similarly, identification of LAR tumors has highlighted the potential of androgen receptor antagonists, while MES-like subtypes may benefit from inhibitors targeting EMT or stemness-related pathways [24,35]. The translational challenge lies in moving beyond descriptive molecular categorizations toward functional taxonomies that align directly with therapeutic vulnerabilities [9].

Despite significant progress, challenges remain. Many of the proposed classifications, especially for TNBC, are derived from transcriptomic analyses that are not routinely available in clinical practice [36]. Immunohistochemistry-based surrogate panels, incorporating markers have been proposed but require broader validation. Moreover, discrepancies between gene expression and protein abundance limit the reliability of RNA-based classifiers, emphasizing the need for proteomic and functional approaches [29]. Equally pressing is the need to account for temporal heterogeneity: tumors may evolve during therapy, leading to shifts in subtype identity and emergent drug resistance [17]. Thus, static baseline classification may not be sufficient; dynamic monitoring through repeat biopsies or liquid biopsies could provide more accurate guidance [22,24].

Collectively, the limitations of the classical four-subtype model, together with growing evidence of functional heterogeneity, underscore the need to redefine BC taxonomy. Newly recognized subpopulations, such as HER2-low, claudin-low, and refined TNBC subsets illustrate how more granular classification enhances prognostic accuracy and reveals novel therapeutic opportunities [21,34,35]. Advances in omics-based technologies, liquid biopsy biomarkers, and microenvironmental profiling are driving the development of an integrated framework for BC subtyping that transcends static receptor status [31]. Importantly, this evolution in classification is not merely academic but has direct clinical relevance, particularly for aggressive entities, such as TNBC, where conventional paradigms have proven inadequate [24,29]. The purpose of this review is to synthesize current insights into emerging BC subpopulations, emphasizing their biological distinctiveness, prognostic relevance, and therapeutic potential. By critically appraising novel molecular and functional classification systems, we aim to highlight pathways toward more precise, personalized, and effective management of BC, moving beyond the constraints of traditional subtyping.

## 2. Classical BC Subtypes and Their Limitations

The classical immunohistochemistry-based subtypes, Luminal A, Luminal B, HER2-enriched, and TNBC/basal-like—continue to serve as a practical framework in clinical BC management [2]. However, their ability to represent the true biological spectrum of the disease is increasingly limited [14]. Originally introduced as practical clinical stand-ins for intrinsic molecular subtypes, these categories encompass only a fraction of the genomic and phenotypic heterogeneity that influences tumor behavior and treatment response, a topic discussed in detail in the preceding section [37].

Multiple studies have shown considerable discordance between IHC-defined subtypes and gene-expression profiles [38]. This inconsistency reflects more than methodological variation that arises from fundamental biological differences, including post-transcriptional regulation, variation in protein turnover, and spatial heterogeneity within the tumor itself [39]. As a result, tumors with the same IHC classification may operate under distinct transcriptional programs, leading to divergent growth dynamics, immune interactions, and vulnerabilities to treatment, features that remain invisible when relying solely on ER, PR, HER2, and Ki-67 assessment [40,41].

These limitations have clear clinical consequences. Patients assigned to the same subtype often display markedly different risks of relapse, metastatic patterns, and responses to endocrine therapy, chemotherapy, or targeted agents [14]. Static biomarker evaluation provides only a snapshot of a tumor that is constantly evolving, and it does not account for clonal competition, microenvironment-driven plasticity, or treatment-induced shifts in biological state [42]. This is particularly evident in tumors with borderline or mixed characteristics, such as ER-positive cancers with very high proliferative indices, where clinical behavior frequently diverges from what the nominal subtype would predict [43]. Moreover, the classical system struggles to incorporate biologically meaningful subgroups that fall between established categories [21,22]. HER2-low cancers provide a clear example: despite historically being grouped as HER2-negative, they exhibit distinct biology and demonstrate differential sensitivity to modern antibody–drug conjugates [44]. Similarly, tumors expressing basal cytokeratins despite retaining hormone receptor or HER2 positivity challenge the presumption that basal-like features are synonymous with triple-negative disease, revealing a degree of phenotypic overlap that is not accommodated by current classifications [23]. Taken together, these observations illustrate why the four canonical subtypes, while clinically useful, provide an incomplete representation of BC biology. A more refined, multidimensional approach, integrating molecular, functional, and microenvironmental information, is necessary to improve prognostic accuracy and to more effectively guide personalized therapeutic strategies [21]. A comparative summary of these subtypes is presented in Table 1.

## 3. Emerging Breast Cancer Subpopulations

Although the classical BC subtypes, such as luminal A/B, HER2-enriched, and basal-like, have substantially advanced our understanding of disease biology and guided clinical management, they only partially capture the underlying heterogeneity of BC. Increasingly, complementary classification frameworks have been developed, including histological (morphology-based), molecular (gene expression–driven), functional (cancer stem cell–related), and pathological (immunohistochemical and clinical) approaches [45,46]. Together, these systems underscore additional layers of tumor complexity beyond traditional schemes and have facilitated the recognition of emerging subpopulations with distinctive molecular, functional, or cellular features that refine prognosis and therapeutic stratification [47]. For instance, histological classification distinguishes ductal from lobular carcinoma based on morphology and growth patterns, while molecular classification identifies intrinsic subtypes, such as Luminal A and B or basal-like, according to gene expression profiles. Functional classification emphasizes cancer stem cells (CSCs), defined by markers such as CD44^high^/CD24^low^ and ALDH1, which are associated with aggressiveness and treatment response [47]. Finally, pathological classification integrates immunohistochemical and clinical parameters, including Ki-67, hormone receptor expression, and tumor grade to assess tumor behavior and predict outcome [48]. An overview of these emerging subpopulations and their defining molecular and clinical features is illustrated in Figure 1.

### 3.1. Claudin-Low

Advances in molecular profiling have uncovered BC subgroups that transcend the boundaries of traditional intrinsic classifications. Among these, the claudin-low phenotype was one of the first non-classical entities to be described, originally identified in 2007 as a distinct molecular subtype [49]. Subsequent studies, however, have reframed claudin-low not as a fixed intrinsic category but as a dynamic phenotypic state, often overlapping with other molecular subtypes [50]. Interestingly, claudin low BC can be better understood as a spectrum rather than a binary classification, with tumors varying from those closer matching intrinsic subtypes and those with purely claudin-low characteristics [50]. They account for approximately 8% of invasive BC and most frequently present within the triple-negative spectrum (ER−/PR−/HER2−) [51]. Traditionally, they are associated with poor prognosis but they show an intermediate response to standard preoperative chemotherapy [52]. However, more recent studies suggest that claudin-low status may not be an independent indicator of worse survival outcomes [50].

Regarding the genomic features that define claudin-low, it is characterized by low expression of genes involved in tight junctions and cell–cell adhesion (including claudins 3,4,7, occludin and E-cadherin) [49]. In contrast, claudin-low tumors display high expression of EMT genes and exhibit a stem cell-like, less differentiated transcriptional profile [52]. Immunofluorescent staining has revealed that many claudin-low tumors contain cells that simultaneously express both epithelial (keratin 5/19) and mesenchymal (vimentin) markers [52]. This dual positive expression confirms the transitional EMT state and the fact that mesenchymal features are intrinsic to the tumor cells [52]. When the association with known markers was examined, it was observed that claudin-low tumors had higher probability to express ALDH1 and to exhibit a CD44^high^/CD24^−/low^ phenotype [51].

The ontogeny of claudin-low BC remains unresolved, with three predominant models proposed. One suggests origin from mammary stem cells (MaSCs), another implicates dedifferentiation of luminal epithelial cells via EMT activation; and a third argues for a heterogeneous entity arising through multiple evolutionary trajectories rather than a single progenitor cell type [53]. The third theory views claudin-low BC as a heterogeneous group of tumors that can arise through multiple paths rather than from a single cell type [53,54]. A notable study revealed 3 different subgroups of claudin-low BC that likely originate from distinct origins, such as basal-like BC, luminal BC, and also normal MaSCs [54].

In addition, claudin-low tumors often exhibit a unique structure of TME, with high immune and stromal cell infiltration and high expression of T and B lymphoid cell markers (CD14, CD79a), as well as elevated PD-L1 and IL-6 expression [50,53]. Recent morphological studies further support this, showing significantly elevated levels of TILs both within and around the tumor area, highlighting the importance of further research on claudin-low BC and immunotherapy [55].

### 3.2. Luminal Androgen Receptor (LAR)

Another molecular subtype of TNBC is LAR. It was first identified by Lehmann et al., who classified TNBC into six molecular subtypes, with LAR characterized by AR signaling and luminal gene expression [56]. LAR has since been recognized as a distinct subtype and it occurs in about 15% of TNBC cases [57]. These tumors tend to be lower grade and exhibit a reduced proliferation (low Ki-67), with minimal expression of basal markers CK5/6 and p63, while they have a notably higher expression of AR, FOXA1, FGFR4 and ERBB2-related genes [57,58]. On the other hand, they have low expression of CDK6, BRCAness and p53 markers [57,58]. While these tumors are generally less aggressive, they are often unresponsive to traditional chemotherapy but are potentially responsive to targeted therapies [57]. In support of this, a large meta-analysis showed that LAR patients have significantly reduced pathological complete response (pCR) rates following neoadjuvant chemotherapy compared to non-LAR patients, who were twice as likely to respond [59].

Recent research on regulatory RNAs abnormally expressed in many tumors, has revealed that the LncRNA HOTAIR may stabilize AR protein levels and promote androgen-independent AR signaling, as has been previously observed in prostate cancer models [60,61]. This finding is particularly important, because although anti-androgen therapies appear promising in AR positive TNBC, HOTAIR may contribute to therapy resistance and limited effectiveness, highlighting the need for combination treatment strategies [60]. Additionally, LAR tumors have a more immunosuppressive microenvironment, characterized by lower levels of TILs and decreased immune activity. In contrast, they show an increased presence of myofibroblast-like cancer-associated fibroblasts (CAFs), which may hinder drug delivery and contribute to therapeutic resistance [59].

### 3.3. Basal-like Immune Activated and Immune Suppressed

Basal-like BC can also be further subdivided based on differences in immune system involvement, leading to the identification of two subtypes: BLIA and BLIS. BLIA accounts for approximately 20–30% of TNBC and BLIS 25–40%, making it the most common TNBC subtype [62]. Both of these subtypes are characterized by high chromosomal instability score and enrichment for DNA damage repair processes [62,63,64]. Transcriptomic profiling studies have shown that BLIS subtype is characterized by downregulation of immune-related pathways, including B cell, T cell and NK cell signaling, as well as cytokine pathways [65]. Subsequently these tumors have poor antigen presentation and poor communication with the immune system and are unlikely to respond to immunotherapy. BLIS tumors also express several SOX transcription factors, which are linked to stem cell-like behavior and aggressive growth. This contributed to their generally poor prognosis [62,65,66].

On the contrary, BLIA, the second basal-like subtype of TNBC, shows high activity of immune pathways and tends to have strong immune responses [65]. BLIA tumors are characterized by TP53 mutations and CDK1 amplification. They also exhibit high expression of STAT genes and immune checkpoints like PD-1, PD-L1 and CTLA4 [62,65]. Due to this high immune activity, BLIA is linked to the best prognosis among the basal-like subtypes [65,67]. They also show low activity in stromal and metabolic pathways, which also improves their response to treatment [62]. Therefore, BLIA could prove to be responsive to immune-based therapies such as checkpoint inhibitors and tumor vaccines [68]. As for BLIS tumors, a recent analysis identified CCR5 and IFNG as potential biomarkers linked to relapse-free survival, suggesting that targeting suppressed immune pathways could improve the outcome of BLIS-TNBC [69].

### 3.4. BRCAness

A related category with ties to basal-like tumors is BRCAness, a phenotype that mimics the BRCA1/2 loss without the specific gene mutations [70]. BRCA1 and BRCA2 are key tumor suppressors, and they are essential for homologous recombination repair (HRR) where BRCA1 promotes DNA end processing and BRCA2 assists RAD51 loading for error-free repair of double-strand breaks. Both proteins are also critical for replication fork protection (RFP) by helping to prevent nucleolytic degradation [70]. Since HRR is vital for maintaining genomic integrity by accurately repairing DNA double-strand breaks, mutations in BRCA1/2 compromise this repair pathway and lead to tissue-specific predisposition for breast and ovarian cancer, highlighting the unique roles of these genes in tumor biology [71].

BRCAness may arise from somatic mutations, promoter hypermethylation or loss of function in a variety of HRR-related genes such as PALB2, ATM, CHEK2 and RAD51C [72]. However, BRCAness-related gene alterations are not limited to BC but are prevalent across multiple cancer types [73]. Still, defects in BRCAness genes are considered strong prognostic markers, comparable to BRCA1/2 mutations [73]. This highlights the need for further research on BRCAness and the diagnostic and therapeutic implications beyond traditional BRCA-mutated cancers.

### 3.5. HER2-Low and HER2 Ultra-Low

Another emerging category with clinical relevance is HER2-low BC. Although it is not formally classified as a molecular subtype, HER2-low is recognized as a distinct subgroup within the HER2-negative spectrum. HER2-low BC is defined by a 1+ or a 2+ score in an immunohistochemistry (IHC) assay and negative in situ hybridization (FISH) assay [74,75,76]. It is now believed that approximately 45–55% of tumors that are classified as Her2 fall under the HER2-low category [75,77]. However, HER2-low is not a uniform subgroup, about 60–80% of these tumors are HR positive and only 20–40% are HR negative [75,78]. Her2-low tumors that are HR positive often show characteristics of the luminal subtypes, while Her2-low tumors that are HR negative tend to display basal like molecular features, are more aggressive and often overlap with TNBC [75,77]. Additionally, regarding HR-positive BC, HER2-low tumors were revealed to have higher ERBB2 compared to HER2-zero, suggesting a biological distinction [78].

To further elucidate these molecular differences, a very comprehensive analysis by Berrino and colleagues examined 99 HER2-low tumor samples, comparing their mutation rates and gene expression patterns with those of HER2-negative and HER2-positive [79]. The study identified that HER2-low BC commonly has mutations in PIK3CA, GATA3, TP53 and ERBB2. Also, HER2-low tumors with IHC score 1+ were molecularly similar to HER2-zero (IHC 0), while those with IHC score 2+ (without ERBB2 amplification) resembled HER2-positive tumors and presented more complex profiles. This highlights the heterogeneity within HER2-low BC and its distinct position between HER2-negative and HER2-positive groups [79].

It is evident that the traditional binary HER2 status classification is evolving with the recognition of HER2-low category, and a new concept in the HER2 spectrum—HER2 ultra-low—has also emerged. These are tumors characterized by very minimal HER2 protein expression but no gene amplification, which sets them apart from the classical HER2-positive subtype [80]. HER2 ultra-low presents faint staining in ≤10% of tumor cells in a subset of BC with IHC score-0 [81,82]. A study of Chinese BC patients revealed that HER2 ultra-low tumors showed distinct differences from HER2-low in nodal stage, hormone receptor status, Ki-67 expression, histologic type and treatment patterns [83]. However, HER2 status showed no prognostic value among HER2 negative spectrum tumors [83]. On the contrary, HR positive status was associated with better disease-free survival outcomes and HR positivity (81%) was more frequent in the HER2-ultra-low group [83]. However, further studies are needed to define the HER2-ultra-low category and predictive biomarkers.

### 3.6. ER-Low

Another subgroup worth highlighting is ER-low BC, defined by tumor cells with low levels of estrogen receptors. It was first introduced by the 2020 ASCO/CAP Guideline Update as a new reporting category, where only 1–10% of cells stain positive for ER [84]. Usually, ER expression is bimodal (either no cells are stained or the majority of cells are stained for ER), this makes ER-low atypical [85]. Although it is not officially recognized as a full molecular subtype, it represents a small proportion of BCs (3–9%) and it is increasingly recognized as a distinct clinical case [86]. The low expression of ER may be observed de novo or may arise later as the disease evolves [87,88,89]. There is still a lot of uncertainty revolving around the behavior of ER-low tumors and their response to endocrine therapy [90]. Although ER-low BC is associated with poorer survival outcomes compared to ER-positive, a recent meta-analysis of 12 cohort studies demonstrated that ER-low tumors respond more favorably to neoadjuvant chemotherapy, suggesting a more chemosensitive phenotype [90].

To better understand this behavior, it is important to illuminate the differences between the immunological characteristics of ER-low tumors and both the ER-positive and TNBC. A recent study, utilizing flow cytometry, revealed lower levels of CD4^+^ CD25^+^ activated lymphocytes in ER-low tumors in comparison to TNBC [86]. Interestingly, it was also found that the pCR rate was lower in the ER-low group than in TNBC, a result which is potentially linked to these immunological differences [86]. The complex biology of ER-low tumors is also reflected by their molecular heterogeneity, as they often align with basal-like or HER2-enriched rather than with classical luminal types. It is found that PR status may help better subcategorize ER-low BC, as ER-low/PR positive tumors show more estrogen activity than ER-low/PR negative [91]. Despite these insights, further molecular studies are needed in order to understand the complexity of this uncommon cancer subtype [91].

### 3.7. Tall Cell Carcinoma with Reversed Polarity (TCCRP)

In addition to molecular defined subtypes, recent attention has been drawn to histologic variants including the rare Tall Cell Carcinoma with Reverse Polarity (TCCRP). TCCRP was first described in 2003 as a “BC resembling the tall cell variant of papillary thyroid carcinoma” and it has recently been recognized as a distinct entity in the 5th edition of the WHO classification of BCs [92]. The typical morphology of TCCRP is solid and papillary structures, tall columnar tumor cells with nuclear groove and apically located nuclei (reverse polarity) [93]. TCCRP usually presents as TNBC, being negative for ER, PR, HER2, TTF-1 and thyroglobulin [94]. Despite that, they lack the aggressiveness that is typically associated with TNBC and they are considered as low-invasive tumors [93]. However, a recent case study documented direct skin invasion [95]. Additionally, a separate case report study has revealed a molecular profile of pathogenic mutations of IDH2 and PIK3CA genes, with IDH2 being a diagnostic hallmark [96]. Notably, these tumors express CK7, CK5/6, Calretinin, GATA-3,GCDFP-15 and mitochondrial markers [94]. The Ki-67 proliferation index is typically low (<5%), and myoepithelial markers as p63, Ker5/6 and SMMS are negative [97]. Given the rarity of this subtype, further research is essential to better understand the clinical behavior and optimal management of TCCRP.

Another distinctly recognized histologic group is neuroendocrine (NE) BC, which is very heterogeneous. NE differentiation of BC refers to the presence of architectural and cytologic features that resemble NE tumors of the lung or gastrointestinal track, as well as the expression of NE markers such as chromogranin (CG) and synaptophysin (SYN) in more than 50% of cell populations [98]. Recently, Insulinoma-associated Protein 1 (INSM1) has also been suggested as a strong marker [98]. Nevertheless, BCs with both NE morphology and NE marker expression are quite rare, whereas infiltrating BCs of no special type (IBC-NST) that display only NE marker expression, without NE morphology, are more common [98].

The 5th WHO classification has aligned the categorization of neuroendocrine neoplasms of the breast (Br-NENs) with that of NENs in other organs, distinguishing Br-NENs into well-differentiated neuroendocrine tumors (NETs, low grade) and poorly differentiated neuroendocrine carcinomas (NECs, high grade). Despite this classification, considerable overlap remains between NE-differentiated BCs and invasive BC of no special type (IBC-NST), which often complicates diagnosis [98,99]. Furthermore, NECs may be subdivided further into small cell neuroendocrine carcinoma (SCNEC) and large cell neuroendocrine carcinoma (LCNEC) [99]. A recent study identified co-alterations of TP53 and RB1 in 86% of SCNEC and in 50% of LCNEC, while 100% of the SCNEC and 50% of the LCNEC cases that were studied showed RB loss [100]. To conclude, although uncommon and heterogeneous, BCs with NE differentiation typically exhibit HR positive and HER2 negative immunohistochemical profiles, suggesting that hormonal therapy and CDK4/6 inhibitors could be appropriate treatment options [99].

### 3.8. New Diagnostic and Classifying Techniques

With all of the above in mind, given the high diversity of BC, the existing subtyping schemes and diagnostic panels cannot fully capture the differences between each type. A recent study presented a new method to classify BC samples, based on genome-wide metabolic gene expression patterns [101]. This new classification system combines machine-learning clustering methods to group the samples and categorizes the samples into four classes with distinct metabolic and immune profiles [101]. In addition, DNA methylation has been explored as a tool for BC subtyping, and while promising, efforts to define new subtypes have been somewhat inconsistent across studies [102]. However, recent advances in using methylation data to assess tumor composition may contribute to future improvements in BC classification and more personalized treatment approaches. Other genetic profiling studies, such as investigating missense variants in *PRKCQ,* have begun to reveal SNP-related associations with established subtypes (like Luminal A and HER2-positive), as well as with disease stage and BRCA status. While this does not reveal entirely new subtypes, such findings suggest the potential of integrating SNP genotyping into personalized risk assessment and subtype refinement [103].

Emerging diagnostic panels using CTCs may offer valuable insights into the heterogeneity of BC subtypes. By capturing live tumor cells from the blood stream, these panels reflect molecular changes and clonal evolution that tissue biopsies and ctDNA might be missed. Even though technical challenges remain, CTC-based diagnostics are a very promising approach for identifying targets and resistance mechanisms in diverse BC subtypes, improving personalized treatment [104]. In particular, transcriptomic profiling of CTCs may offer valuable insights into metastatic mechanisms, tumor heterogeneity, while also serving as a method for uncovering emerging biomarkers in the least invasive way [105]. Moreover, spatial transcriptomics enable the localization and differentiation of gene expression patterns within defined tissue regions, revealing the spatial organization of tumor cells within their microenvironment [106]. In addition, spatial profiling using the GeoMx Digital Spatial Profiler (DSP) allows quantification of RNA and protein expression in anatomically distinct tumor regions and identification of TME composition [107]. These innovative tools highlight the recent multidimensional diagnostic strategies aiming to characterize BC more accurately. At the core of these advances lies a fundamental challenge: the heterogeneity of BC.

## 4. Functional Heterogeneity Within Breast Cancer Subtypes

As BC research advances, the identification of emerging subtypes continues to refine the classification and therapeutic approaches. Despite this progress, BC exhibits significant biological diversity and variation in treatment response, not only between patients and tumors (intertumor heterogeneity) but also within a single tumor (intratumor heterogeneity) [108]. This heterogeneity arises from variation in genomic, epigenomic, transcriptomic, and proteomic profiles of cancer cells, which then influence important tumor behaviors including growth, apoptosis, metastasis, and therapeutic response [109].

Even within well-defined intrinsic subtypes, considerable differences in clinical outcomes and disease progression persist [108]. Single-nucleus RNA sequencing and spatial transcriptomics have revealed distinct malignant epithelial populations, including cycling luminal-B cells, genomically unstable luminal-A cells, and basal-like subtypes, with some being directly linked to poor prognosis [110]. Another recent study, using multiplex immunofluorescence (MxIF) imaging, revealed that over one third of the cases that were studied had intratumoral spatial heterogeneity. High variability of biomarker expression—especially in luminal A tumors—and differences between HER2 protein and gene expression were observed [111].

To unravel this complexity, it is essential to consider key cellular mechanisms, such as phenotypic plasticity and the role of CSCs. Proliferating tumor cells require nutrient and oxygen availability for biosynthesis of macromolecules, whereas migrating or circulating tumor cells prioritize metabolic pathways that generate more ATP, to support motility and survival [112,113]. This metabolic flexibility is closely linked to phenotypic plasticity. Metabolic regulation plays a crucial role in the transitions between E/M states, particularly through epigenetic modifications such as methylation and acetylation, which support phenotypic remodeling [112]. In aggressive forms of BC, including TNBC types, phenotypic plasticity—driven by EMT and its reverse process (MET)- plays a key role in intratumoral heterogeneity and metastatic progression. Cancer cells often exist in hybrid E/M states, that enhance their ability to migrate, their invasiveness, and adaptability factors that complicate therapeutic targeting and contribute to poor prognosis [114]. Moreover, BC cells can dynamically switch between glycolysis and oxidative phosphorylation in response to fluctuating microenvironment conditions, such as oxygen or nutrient availability. This metabolic heterogeneity promotes survival under stress and result in resistance to metabolic-targeted therapies [115].

BCSCs represent another major factor of intratumoral heterogeneity due to their self-renewal ability, plasticity and resistance to conventional therapies [116]. Evidence suggests that the proportion of BCSCs increases following chemotherapy, thereby promoting resistance and facilitating tumor recurrence [116]. Importantly, a study on endocrine-resistant metastatic BC provided evidence that CAF-derived microvesicles carrying miR-221 promote the conversion of ER^high^ tumor cells into CD133^high^/ER^low^ BCSCs, supporting the concept that microvesicle-mediated miRNA transfer can convert non-stem cancer cells into therapy resistant CSCs [117].

This dynamic plasticity of BCSCs is also closely related to the concept of clonal evolution. According to this model, cancer cells within a tumor gradually acquire random mutations, giving some cells a growth advantage and promoting aggressive tumor progression [118]. Large-scale analyses have shown that mutations in key genes such as PIK3CA, TP53 and MYC can arise early or later during tumor progression and that many of these mutations are heterogeneous within the same tumor [119]. Such intratumoral heterogeneity poses a significant challenge for biomarker analysis and limits the ability of conventional tissue biopsies to predict treatment response [119]. As an alternative, ctDNA-based liquid biopsy has gained attention as an important non-invasive tool, that may reflect metastatic clones more accurately than primary tumor biopsies [120].

In addition to genetic and epigenetic diversity within tumor cells, the TME plays an important role in functional heterogeneity. Composing of tumor cells as well as endothelial cells, fibroblasts, immune cells, and ECM components, the TME forms in response to metabolic demands of rapidly growing cancer cells. Through biochemical and biomechanical signaling, tumor cells actively manipulate the surroundings and create a complex network that contributes to intratumoral diversity [121]. While tumor cells initiate TME formation, TME also actively shapes tumor behavior. Environmental cues from immune cells, CAFs, and EMC components induce genetic and epigenetic changes that enhance heterogeneity [121]. For example hypoxia-driven HIF-1 activation, CAF signaling and ECM remodeling, drive phenotypic changes and therapy resistance [121]. More specifically, various TME cells, including M2 macrophages, myeloid-derived suppressor cells, CAFs, and endothelial cells, work together to support cancer cell survival, suppress immune responses and alter tumor structure, thus reducing the effectiveness of treatments [122]. Interestingly, TME itself is highly heterogeneous, not only across patients but between subtypes. For instance, in TNBC the microenvironment varies significantly, with some subtypes showing immune-inflamed phenotypes, while others display immune-suppressive or immune-cold profiles [123]. Taken together, both tumor cell heterogeneity and the complex TME are pivotal in shaping tumor behavior, influencing prognosis, and driving disease progression (Figure 2).

In addition, tumor heterogeneity may also evolve in response to selective pressure such as therapy. Transcriptional and epigenetic changes, often lead to the emergence of treatment resistant clones and facilitate metastatic progression [119]. Recent advances in single-cell RNA and ATAC sequencing have allowed detailed analysis of these transcriptional and epigenetic changes in tamoxifen-resistant tumors [124]. These studies identified distinct cancer cell states specific to resistant phenotypes, revealing dynamic changes in chromatin accessibility and gene expression associated with therapy escape. Of particular note, BMP7 has been shown to promote tamoxifen resistance via MAPK signaling [124]. In general, resistance may be present at diagnosis (primary) or arise through treatment-driven selective pressure (acquired), with mechanisms influenced by drug potency and binding affinity [125]. These findings underscore the essential role of accurate prognostic and predictive biomarkers to assist in tailoring personalized treatments for aggressive tumor subpopulations.

## 5. Prognostic and Predictive Biomarkers in Aggressive Subpopulations

The clinical trajectory of BC reflects the canonical molecular subtypes while simultaneously being shaped by the emergence of discrete subpopulations with distinct functional and biological attributes [126]. Among these, HER2-low, claudin-low, and BRCA-deficient or “BRCAness” tumors exemplify how unique biomarker landscapes may refine prognosis and guide therapy beyond conventional classification [127]. Within this framework, biomarkers serve dual and complementary purposes: they identify patients at higher risk of relapse (prognostic) and predict the likelihood of therapeutic benefit or resistance (predictive) [128]. A central question, therefore, is how established and emerging biomarkers perform across these aggressive subsets. The diversity of prognostic and predictive biomarkers shaping these aggressive breast cancer subpopulations is summarized in Figure 3, providing a visual framework for the sections that follow.

### 5.1. Proliferation and Genetic Drivers

Proliferation-related proteins represent one of the earliest and most widely studied categories of biomarkers in BC. Ki-67, in particular, has long been used as a surrogate of tumor growth kinetics and predictor of response to neoadjuvant chemotherapy (NACT) [129]. Although interpretation remains complicated by technical variability and inconsistent cut-off thresholds, elevated Ki-67 consistently correlates with worse survival in high-grade tumors [130]. Recent clinical trials have even incorporated Ki-67 into treatment algorithms, exemplified by the approval of adjuvant abemaciclib for patients with Ki-67 ≥ 20% [129]. Importantly, its prognostic significance is context-dependent: while luminal tumors rely heavily on Ki-67 thresholds for risk stratification, basal-like and claudin-low cancers, already defined by high proliferative indices, derive little incremental value from its measurement [131].

Beyond proliferative markers, genetic drivers, exemplified by TP53 mutations, further underscore the complexity of biomarker interpretation. TP53 is mutated in up to 80% of TNBCs, and although dysfunction in this pathway promotes genomic instability and poor prognosis, the clinical implications are heterogeneous [28]. Certain mutations confer chemosensitivity, while others predict aggressive relapses, emphasizing the need for nuanced interpretation. Importantly, novel therapeutic strategies designed to restore p53 function may reposition TP53 from a primarily prognostic biomarker to a therapeutic vulnerability [132]. Additional proteins, including EGFR, FOXM1, and matrix metalloproteinases (MMPs), reinforce the biology of aggressive disease: EGFR is commonly overexpressed in basal-like BCs, FOXM1 deregulates cell cycle progression, and MMPs facilitate invasion and metastasis [28,133]. Collectively, these alterations converge on pathways that sustain tumor aggressiveness and poor outcomes.

In parallel, the IGF-1/IGF-1R signaling axis has been increasingly recognized as subtype-specific in BC [134]. Elevated IGF-1 and IGF-1R expressions correlate with poor prognosis, early recurrence, and resistance to standard therapies [2,135]. Their functional interplay with HER2 and EGFR pathways, together with marked enrichment in TNBC and claudin-low tumors, underscores the axis as a marker of aggressive disease [136]. In TNBC, in particular, ligand–receptor activation and downstream signaling via PI3K/AKT and MAPK cascades have been linked to tumor growth, invasion, and therapy resistance [137]. Recent studies highlight not only its biological relevance, but also its therapeutic potential, with combinatorial approaches targeting IGF-1/IGF-1R alongside other oncogenic drivers showing encouraging results [134,135].

At the genomic level, mutations in signaling pathways add further complexity. PIK3CA mutations exemplify this challenge, being frequent in luminal tumors but associated with divergent prognostic effects depending on molecular context [138]. In HER2-positive cancers, certain variants correlate with improved disease-free survival, whereas others, such as H1047R, are linked to adverse prognosis. These inconsistencies underscore the importance of subtype-specific interpretation rather than broad generalizations [129].

### 5.2. Homologous Recombination Deficiency and BRCAness

Homologous recombination deficiency (HRD) represents one of the most clinically actionable biomarker landscapes in BC, particularly within the TNBC subtype [139]. Germline mutations in BRCA1 and BRCA2, historically associated with universally adverse outcomes, have undergone a conceptual reappraisal in recent years [140]. Rather than serving solely as markers of poor prognosis, these alterations now define tumors with specific therapeutic vulnerabilities, most notably sensitivity to platinum-based chemotherapy and poly (ADP-ribose) polymerase (PARP) inhibition [28]. This shift illustrates how classical prognostic indicators may evolve into predictive biomarkers once effective targeted strategies are available [141]. Beyond BRCA1/2, the spectrum of HRD extends to additional components of the DNA repair machinery, including PALB2, RAD51C/D, BARD1, ATM, ATR, CHEK1/2, and WEE1 [139]. Alterations within these genes disrupt the integrity of the homologous recombination pathway, generate genomic instability, and create a phenotype broadly termed “BRCAness” [126]. Importantly, incorporating these extended alterations into clinical testing frameworks enhances the sensitivity of HRD detection and allows for more precise stratification of patients who may benefit from DNA-damaging agents [142]. Collectively, this expanding landscape underscores the dual role of HRD-related biomarkers as both indicators of intrinsic tumor biology and as practical tools for guiding therapeutic decision-making in aggressive BC subpopulations [140].

### 5.3. Epigenetic and Non-Coding RNA Biomarkers

Epigenetic mechanisms add yet another dimension to biomarker discovery. DNA methylation patterns affecting tumor suppressors, exemplified by BRCA1, RASSF1, and PITX2, have consistently been associated with aggressive behavior and poor outcomes in ER-positive and HER2-negative tumors [138]. Interestingly, PITX2 methylation appears subtype-specific, predicting unfavorable outcomes in luminal cancers but conferring increased chemotherapy sensitivity in TNBC [143]. Similarly, methylation of ESR1 promoter and enhancer regions has been implicated in endocrine resistance, with detection in cfDNA offering a minimally invasive strategy to monitor resistance and adapt therapy accordingly [144].

In addition to epigenetic regulation, post-transcriptional mechanisms significantly shape tumor behavior. MiRNAs constitute a key layer of regulation: upregulation of miR-21, miR-27a, and miR-210 has been associated with poor survival in TNBC, whereas downregulation of miR-155 correlates with inferior outcomes [28]. Claudin-low tumors are characterized by loss of the miR-200 family, consistent with their stem-like and mesenchymal phenotype [145]. Restoration of miR-200c suppresses claudin-low growth, underscoring its mechanistic role [146]. Emerging evidence points to circular RNAs (circRNAs) as regulators of miRNA activity. For example, circGFRA1 promotes proliferation in TNBC by sequestering miR-34a, and high levels correlate with poor histological grade [147]. Viewed in concert, these epigenetic regulators underscore the complexity of biomarker biology across aggressive subgroups.

### 5.4. Proteomic and Immune-Related Biomarkers

Proteomic biomarkers expand the spectrum beyond classical receptors. A prominent example is HER2-low disease, which was previously categorized under HER2-negative tumors but is now recognized as clinically relevant due to responsiveness to antibody–drug conjugates such as trastuzumab deruxtecan [148]. Intratumoral HER2 heterogeneity, even within HER2-zero tumors, may partly explain why certain “ultralow” cases demonstrate benefit from ADC therapy [35]. Moreover, HER2 mutations without amplification, frequently enriched in HER2-low tumors, emerge as predictive markers of response to tyrosine kinase inhibitors (TKIs), such as neratinib [148].

Beyond HER2-driven phenotypes, claudin-low tumors exemplify another proteomically distinct subgroup. These are characterized by high expression of EMT-related proteins and inflammatory cytokines, including IL-6 and VEGF-C [145]. The inflammatory profile is further reinforced by CXCL8/IL-8–driven signatures that sustain an immunosuppressive microenvironment and, at the same time, serve as predictors of metastatic progression [145]. Importantly, evidence of MEK pathway activation in this subtype highlights potential therapeutic vulnerabilities, with phospho-MEK and phospho-ERK proposed as functional biomarkers of sensitivity to MEK inhibition [149].

Immune checkpoints have redefined the landscape of biomarker discovery. PD-L1 expression is enriched in TNBC and correlates with poor outcomes, yet it also identifies subsets of patients likely to benefit from checkpoint blockade [150]. Nevertheless, large clinical trials such as IMpassion130 and KEYNOTE-355 have demonstrated that PD-L1 is not universally predictive, as some PD-L1–negative tumors still respond [151]. This inconsistency has stimulated exploration of additional checkpoints, including LAG-3, TIGIT, and TIM-3, which are frequently co-expressed with PD-1/PD-L1 and may provide a more comprehensive understanding of immune escape mechanisms [28,151].

Complementing immune checkpoint biomarkers, TILs represent validated prognostic indicators, particularly in TNBC and HER2+ disease, where higher density is associated with improved survival and enhanced chemotherapy response [152]. Beyond overall density, the composition of TIL subsets most notably CD8+ cytotoxic T cells and the presence of tertiary lymphoid structures (TLSs) strongly predicts immunotherapy responsiveness [153]. Epigenetic immune signatures, such as MeTIL profiles, further refine quantification of immune infiltration and may contribute to patient stratification [143].

Parallel advances have established additional protein-based biomarkers with predictive significance. TROP-2 has emerged as a key target in TNBC and HER2-low cancers, where sacituzumab govitecan improves survival in heavily pretreated patients [154]. At the same time, rare but actionable genomic events such as NTRK fusions and FGFR1/2 amplifications, though infrequent, underscore the importance of broad molecular profiling to identify therapeutic opportunities in aggressive disease [155].

### 5.5. Liquid Biopsy and Integrated Approaches

Non-invasive approaches are increasingly shaping biomarker discovery. CtDNA offers dynamic insights into tumor burden, resistance mechanisms, and minimal residual disease, often rising before radiologic progression [128]. Specific alterations, including TP53 and ESR1 mutations, are associated with therapeutic resistance [156]. Closely related, cell-free DNA (cfDNA) derived from both malignant and normal cells might reveal tumor-associated mutations such as TP53 and PIK3CA through advanced sequencing and PCR methods [157]. Elevated cfDNA levels correlate with tumor burden and poor survival, while longitudinal changes may signal therapeutic resistance earlier than imaging [128]. Advanced approaches, including cfDNA fragmentomics that interrogate fragment length and termini, alongside methylation-based assays (e.g., mDETECT, cMethDNA), are actively under evaluation as tools for detecting minimal residual disease in TNBC [28].

CTCs provide complementary information, with elevated counts correlating with metastatic spread and adverse outcomes [22]. Beyond their association with metastatic burden, CTCs are increasingly recognized as subtype-informative, with EMT-like and stem-like phenotypes particularly enriched in claudin-low and TNBC subsets, thereby linking their biology to aggressive, poorly classified disease entities [158]. Importantly, phenotypic characterization, such as EMT-like CTCs, CTC clusters, and PD-L1–positive CTCs, yields predictive insights into immune evasion and therapeutic resistance [159]. Similarly, exosomes serve as carriers of oncogenic miRNAs and proteins that mirror the molecular and immune-modulatory status of the tumor [128]. Distinct exosomal miRNA signatures, particularly in TNBC and claudin-low cancers, emphasize pathways of EMT and immune modulation, further supporting their role as biomarkers. Interestingly, in HER2-low tumors, circulating assays may even distinguish ultralow expressors who remain candidates for ADC therapy [148].

The clinical relevance of these biomarkers varies across molecular subgroups. In HER2-low tumors, HER2 expression once deemed insignificant, now determines eligibility for trastuzumab deruxtecan, with “ultralow” tumors also showing potential benefit [148]. ERBB2 mutations without amplification provide an additional layer of predictive information for HER2-targeted tyrosine kinase inhibitors. Within claudin-low BCs, loss of the miR-200 family and upregulation of IL-6 and VEGF-C highlight the centrality of EMT and immune modulation [160]. The presence of IL-8-driven inflammatory profiles and MEK pathway activation further refine this subgroup, supporting exploration of targeted therapeutic strategies. In BRCAness-associated TNBC, BRCA1 promoter methylation and homologous recombination deficiency (HRD) scores aid in identifying patients likely to benefit from platinum chemotherapy and PARP inhibitors [28,143]. Incorporating alterations in a broader set of HR-related genes enhances the sensitivity of HRD detection and improves patient selection.

Given the complexity and heterogeneity of BC, integrated approaches are increasingly essential. Multigene assays such as Oncotype DX and MammaPrint already guide therapeutic decision-making in luminal tumors [161]. For aggressive subtypes, composite biomarker panels that integrate genetic (BRCA1/2, PIK3CA), epigenetic (DNA methylation, miRNAs), proteomic (HER2-low, PD-L1, TROP-2), and circulating markers (ctDNA, cfDNA, exosomes, CTCs) appear most promising [128,156]. For example, combining TIL subset composition with PD-L1 expression and cfDNA dynamics may refine immunotherapy selection in TNBC, while DNA methylation profiling of BRCA1, PITX2, and ESR1, integrated with cfDNA analysis, may inform treatment strategies in more heterogeneous populations [143]. Finally, emerging technologies such as single-cell RNA sequencing and spatial transcriptomics provide unprecedented resolution of intratumoral heterogeneity. These approaches uncover functional niches, including androgen receptor–driven TNBC (LAR subtype) and immune-excluded claudin-low cancers, thereby expanding the repertoire of actionable biomarkers and revealing novel therapeutic vulnerabilities [145,162].

**Figure 3 ijms-26-11599-f003:**
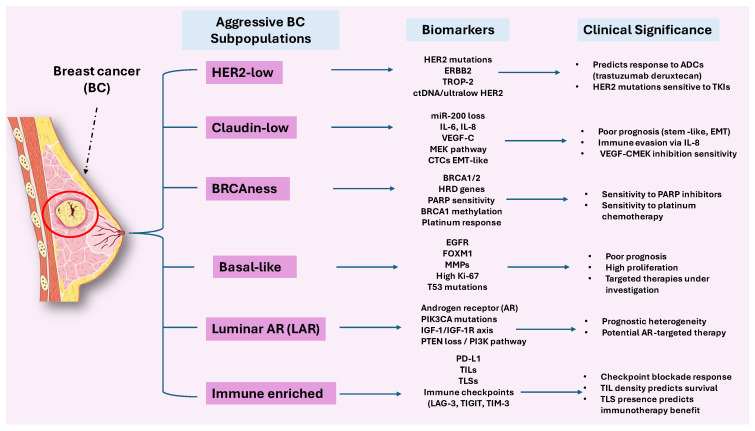
“Prognostic and predictive biomarkers across aggressive breast cancer subsets”: The illustration depicts the main aggressive breast cancer (BC) subpopulations, including HER2-low, claudin-low, BRCAness, basal-like, luminal AR (LAR), and immune-enriched. Each group is defined by a characteristic set of biomarkers, spanning genetic, epigenetic, proteomic, and immune-related alterations. These features not only clarify biological heterogeneity but also carry prognostic weight and guide therapeutic strategies, including antibody–drug conjugates, PARP inhibition, checkpoint blockade, and AR-targeted therapy. By integrating biomarkers with clinical outcomes, the figure highlights how precise molecular profiling may support more tailored treatment decisions in aggressive BC.

### 5.6. Merging Nanoscale and Computational Biomarker Technologies

Advances in nanoscale biophysics, computational analytics, and engineered microenvironments are reshaping biomarker discovery for aggressive BC [163]. High-resolution mechanical imaging now captures subtle changes in cell stiffness, elasticity, and viscoelastic behavior that reliably distinguish malignant from normal epithelial cells and often emerge before detectable morphological alterations, providing an additional layer to classical molecular profiling [164]. At the same time, machine-learning models capable of integrating multi-omics, imaging, and clinical datasets enhance prognostic precision and reveal hidden patterns linked to therapeutic response in heterogeneous subtypes such as TNBC, claudin-low, and HER2-low tumors [165]. Complementing these approaches, biofabricated 3D matrices, constructed through granular suspension systems and drop-on-demand bioprinting, allow BC and stromal cells to be positioned within tunable microporous architectures that mimic healthy or fibrotic microenvironments, enabling controlled study of matrix–cell crosstalk [163,164,165]. Together, these nanoscale, computational, and microengineered technologies broaden the biomarker landscape, offering integrated tools for more accurate risk stratification and individualized treatment planning in aggressive BC.

## 6. Therapeutic Implications and Ongoing Clinical Trials

### 6.1. Targeted Therapies and Immunotherapy

The current cornerstones of BC management remain endocrine therapy, cytotoxic chemotherapy, and HER2-targeted agents [166]. As novel subtypes of BC have gained recognition, the therapeutic landscape has expanded to include targeted agents and immunotherapies [167]. Among the molecularly defined groups now considered clinically actionable are HER2-low tumors and AR-positive cancers [168]. Immunotherapy, in contrast, has shown the greatest efficacy in TNBC characterized by high levels of immune infiltration or PD-L1 expression [169].

Antibody–drug conjugates (ADCs) directed against HER2-low tumors illustrate the practical application of precision oncology [170]. Unlike HER2-amplified cancers, HER2-low tumors exhibit low surface HER2 expression without amplification, historically placing them in the HER2-negative category and excluding them from HER2-targeted treatment [171]. ADCs overcome this barrier by coupling a cytotoxic payload to a HER2-targeting monoclonal antibody, thereby selectively delivering chemotherapy to tumor cells with even modest HER2 expression [172]. Clinical trials have demonstrated that these agents substantially improve progression-free survival in patients with metastatic disease [173]. The clinical success of ADCs has redefined HER2-low BC as a distinct therapeutic entity, prompting ongoing efforts to refine patient selection and optimize treatment combinations, including integration with immunotherapy [174].

AR-positive BCs, particularly within the TNBC subgroup, represent another emerging therapeutic niche. AR signaling has been implicated in tumor growth and survival, making it a promising target for intervention Agents such as enzalutamide and bicalutamide, originally developed for prostate cancer, have demonstrated activity in AR-positive BCs [175]. They may be used either as monotherapies or in combination with other systemic agents [176]. Early-phase studies suggest that AR blockade can induce tumor shrinkage and disease stabilization, offering an alternative strategy for patients lacking other targeted options [177]. Ongoing trials are increasingly focused on combining AR-directed therapy with HER2 inhibitors or immune checkpoint blockade, reflecting a broader shift toward multimodal targeted strategies [178].

Immunotherapy has also emerged as a key modality in BC, particularly when tumors are defined by their immune microenvironment rather than classical molecular markers. In TNBC, the presence of TILs correlates strongly with favorable prognosis and enhanced responses to immune checkpoint inhibitors [179]. Similarly, PD-L1 expression on tumor or immune cells serves as a predictive biomarker for the efficacy of PD-1/PD-L1 inhibitors, especially in advanced TNBC [180]. Combining checkpoint blockade with chemotherapy has significantly extended progression-free survival in PD-L1–positive advanced TNBC, establishing immunotherapy as an important standard-of-care option for this subgroup [181].

An especially promising avenue involves combining targeted therapies with immunotherapy. ADCs not only deliver cytotoxic drugs but also enhance antitumor immunity by promoting immunogenic cell death [182]. Similarly, AR inhibition may alter the tumor immune milieu by shifting the balance between immune-supportive and immune-suppressive cell populations, thereby augmenting the effectiveness of checkpoint blockade [183]. Based on these findings, clinical trials are actively exploring sequential and concurrent administration of targeted agents and immunotherapies to maximize clinical benefit [183].

### 6.2. Stem-like and EMT-Driven Tumors, and Epigenetic Combinatorial Strategies

Beyond conventional genetic targets, growing attention has been directed toward tumors exhibiting stem-like properties or undergoing EMT [184]. CSCs, although representing a minor subpopulation, possess the capacity for self-renewal, differentiation, and tumor initiation. They are strongly associated with therapeutic resistance and metastatic spread [185]. Likewise, the EMT process endows epithelial tumor cells with mesenchymal features, thereby increasing invasiveness, promoting treatment resistance, and facilitating immune evasion [186]. Recognizing these functional phenotypes is crucial for addressing drug resistance and achieving more durable clinical responses [187]. Ongoing clinical trials targeting these functional subpopulations are summarized in Table 2 and Figure 4.

Therapeutic strategies targeting CSCs and EMT pathways primarily focus on key signaling cascades, including TGF-β, Wnt/β-catenin, Notch, and Hedgehog [188]. Early-phase clinical trials have evaluated inhibitors of these pathways as monotherapies or in combination with chemotherapy [189,190]. Preclinical studies suggest that EMT inhibition can reduce cellular plasticity, suppress metastatic potential, and restore chemosensitivity, thereby providing a strong biological rationale for ongoing clinical investigations [191,192].

Epigenetic regulation has also emerged as a promising therapeutic avenue for both stem-like and immune-resistant tumors. DNA methyltransferase inhibitors (DNMTi) and histone deacetylase inhibitors (HDACi) can remodel chromatin structure, reactivate silenced tumor suppressor genes, and enhance tumor immunogenicity [193,194]. These effects simultaneously promote immune-mediated clearance and disrupt CSC and EMT programs. Increasingly, epigenetic modulators are being tested in combination with checkpoint blockade, with the aim of reprogramming the TME from “cold” to “hot,” thereby facilitating immune attack [195].

Ongoing clinical trials are systematically evaluating combinations of functional targeting and immunotherapy, including CSC-directed agents, epigenetic modifiers, and EMT pathway inhibitors [196,197]. Most frequently, these regimens are paired with PD-1 or PD-L1 checkpoint blockade to maximize therapeutic efficacy [198]. The integration of genomic and functional profiling to identify patients enriched for CSC or EMT signatures is becoming increasingly important, ensuring that therapies are precisely directed toward clinically relevant subpopulations [199].

**Table 2 ijms-26-11599-t002:** Ongoing Clinical Trials for Stem-Like, EMT, and Epigenetically Modulated Tumors.

Subpopulation	Targeted Strategy	Trial Phase/Status
CSC-High/EMT-High [190]	TGF-β, Notch, Wnt pathway inhibitors ± chemotherapy	Early-phase/Ongoing
Epigenetically Modulated [194]	HDACi or DNMTi ± immune checkpoint inhibitors	Early-phase/Ongoing
EMT-Prominent TNBC [197]	EMT pathway inhibition ± PD-1/PD-L1 blockade	Early-phase/Ongoing
Stem-Like TNBC [192]	CSC-targeted therapy ± chemotherapy	Early-phase/Ongoing
Combination Approaches [200]	Epigenetic modulation + ADC or checkpoint inhibitors	Early-phase/Ongoing

## 7. Challenges and Future Directions

Despite significant progress, several barriers remain before precision oncology can be fully realized across all BC subtypes, particularly in the identification and management of emerging subpopulations. The intrinsic complexity of tumor biology, including both intertumoral and intratumoral heterogeneity, continues to challenge effective classification and therapeutic targeting [201]. Accurate tumor characterization is often limited by technical constraints, small sample sizes, and the dynamic nature of tumor ecosystems, making it difficult to establish universally applicable treatment paradigms [202].

Detecting and characterizing these subgroups is complicated by spatial and temporal variation in tumor biology. Distinct tumor regions may harbor divergent genetic or phenotypic profiles, and these features may evolve during disease progression or under therapeutic pressure [203]. Standard biopsy approaches capture only a fraction of tumor heterogeneity, often missing CSC-enriched niches, EMT-driven regions, or immune-infiltrated compartments [204]. Such sampling bias risks misclassification, leading to inappropriate therapeutic decisions and inaccurate prognostic assessment [205]. Superficial or limited biopsies may fail to capture deeper invasive fronts or critical microenvironmental elements essential for metastasis and immune escape, reducing the clinical utility of conventional histopathological and molecular testing [206].

Overcoming these challenges requires a paradigm shift in how tumors are defined and stratified, moving from static, morphology-based assessments toward dynamic characterizations that incorporate pathway activity, epigenetic states, and cellular interactions within the microenvironment [207]. Functional mapping approaches can reveal clinically relevant subpopulations that strongly influence therapeutic responsiveness, metastatic potential, and immune evasion [208]. This enables more precise treatment planning and the development of refined prognostic models that better reflect underlying tumor biology [209].

Technological innovations have transformed the field of tumor profiling. Single-cell transcriptomics allows unprecedented resolution of cellular composition, uncovering rare cell populations and functional states not detectable through bulk analysis [210]. When combined with spatial transcriptomics, these methods preserve tissue architecture and provide insight into the spatial dynamics of tumor, stromal, immune, and vascular compartments [211]. Machine learning and artificial intelligence (AI) further enhance these efforts by integrating large-scale datasets to identify hidden patterns, predict clinical trajectories, and anticipate tumor evolution [212]. These technologies have the capacity to significantly improve therapeutic decision-making and patient outcomes. However, their integration into routine clinical practice demands the establishment of standardized methodologies, reliable bioinformatics frameworks, and validation through large prospective studies. At the same time, ethical and practical issues, including tissue availability, data security, and the accelerating pace of technological innovation, must be carefully managed to ensure their equitable and sustainable adoption. Building on these challenges, a growing area of interest involves the incorporation of next-generation immunotherapeutic strategies into the management of aggressive BC subgroups. Early-phase studies evaluating CAR-T and CAR-NK cell therapies targeting HER2, TROP-2, and other lineage-specific antigens have shown preliminary activity, suggesting that engineered cellular therapies may eventually complement current antibody-based and checkpoint-directed approaches [213]. Continued refinement of antigen selection, strategies to overcome the immunosuppressive TME, and improvements in safety engineering will be essential for their broader application [214]. Additionally, integrating spatial multi-omics with AI-driven modeling could enable real-time prediction of emerging resistance pathways, paving the way for adaptive treatment strategies that evolve alongside tumor biology [214,215].

## 8. Discussion

The recognition of emerging BC subpopulations has fundamentally reshaped our understanding of tumor heterogeneity beyond the canonical luminal, HER2-enriched, and TNBC categories [216]. Advances in multi-omics technologies, liquid biopsy approaches, and integrative bioinformatics have revealed that subsets such as HER2-low, claudin-low, BRCA-deficient tumors, and distinct TNBC subtypes represent clinically relevant entities with unique biology and therapeutic vulnerabilities [3,26]. These discoveries challenge the conventional reliance on receptor-based classification and underscore the importance of refined taxonomies that may improve prognostic precision and inform therapeutic decisions [24].

Among the most notable advances is the recognition of HER2-low tumors. Traditionally grouped as HER2-negative, these tumors demonstrate intermediate HER2 expression and have shown remarkable sensitivity to antibody–drug conjugates, such as trastuzumab deruxtecan, redefining their clinical relevance [21]. Similarly, claudin-low tumors represent a distinct group enriched for EMT, stem-like traits, and immunosuppressive microenvironments, all of which confer aggressive clinical behavior and therapeutic resistance [22,23]. The concept of “BRCAness”, encompassing germline and somatic alterations in homologous recombination repair genes, has also expanded the therapeutic landscape by identifying patients who benefit from platinum agents and PARP inhibitors [24]. Within TNBC, multiple classification systems, ranging from Lehmann’s six-subtype model to Burstein’s four-subtype refinement, have highlighted distinct biological programs, including androgen receptor signaling in LAR tumors, immune activation in BLIA, and mesenchymal pathways in MES, each with potential therapeutic implications [11].

Emerging subpopulations differ at the molecular level while simultaneously exhibiting functional plasticity within tumors. Single-cell analyses have revealed coexistence of proliferative and quiescent clones, cancer stem-like populations, and therapy-resistant subclones that evolve dynamically under treatment pressure [17]. Such ITH complicates prognosis and therapy selection, as different cell states within the same tumor may drive resistance and relapse [16]. TME further contributes to this complexity by shaping immune evasion, angiogenesis, and metastatic potential, emphasizing the need to integrate both tumor-intrinsic and extrinsic features into classification systems [217].

The biomarker landscape in emerging subtypes is increasingly diverse. Proliferation markers, including Ki-67 remain relevant in luminal cancers, while TP53 mutations, EGFR amplification, and FOXM1 overexpression dominate in basal-like and claudin-low tumors [126]. Epigenetic regulators, including BRCA1 methylation and miR-200 loss, highlight subtype-specific vulnerabilities linked to EMT and chemotherapy response [34]. Immunological biomarkers such as PD-L1, TIL density, and emerging checkpoints (LAG-3, TIGIT) have gained traction in TNBC and HER2-low subtypes, where immune signatures strongly influence prognosis and therapy response [151,153]. Complementary to tissue biomarkers, liquid biopsy approaches, including CTCs, ctDNA, and exosomal cargo provide minimally invasive methods for real-time monitoring of tumor evolution and resistance [128,218]. Importantly, these biomarkers are not interchangeable but complementary, each offering unique insights into tumor biology and therapy responsiveness [22].

Despite these advances, challenges remain in translating emerging subtype classifications into routine practice. Lack of consensus across molecular taxonomies, variability in methodologies, and limited availability of high-throughput assays hinder clinical implementation [24]. Moreover, resistance to targeted therapies, including PARP inhibitors and immune checkpoint blockade, remains a significant obstacle, necessitating combinatorial approaches that address parallel resistance pathways [10]. Integrative analyses that combine genomics, transcriptomics, proteomics, metabolomics, and TME profiling are essential to define consensus-driven subtypes that are stable across populations [11]. AI and digital pathology hold promise in refining classification and enabling real-time, cost-effective stratification of patients [142]. Ultimately, longitudinal monitoring through liquid biopsy and spatial transcriptomics may bridge the gap between static classifications and the dynamic reality of tumor evolution [17].

## 9. Conclusions

BC is not fully captured by the classical luminal, HER2-enriched, and TNBC categories. Emerging subpopulations including HER2-low, claudin-low, BRCAness-associated, and molecularly defined TNBC subsets, illustrate the spectrum of functional heterogeneity that drives prognosis and therapy resistance. These subtypes are not merely academic refinements but clinically actionable groups with implications for patient stratification, targeted drug development, and biomarker-driven trial design [22,142]. The identification and characterization of these subgroups highlight the necessity of precision oncology. Recognizing molecular and functionally distinct subsets enables more effective allocation of therapies, such as ADCs in HER2-low, PARP inhibitors in BRCA-deficient tumors, or immunotherapies in immune-active TNBCs [11,21]. Looking ahead, the development of advanced immunotherapies, including CAR-T cells, engineered macrophages, and next-generation bispecific antibodies represents a promising direction for targeting the highly heterogeneous subpopulations described in this review [214,215]. Equally important is the appreciation that heterogeneity operates across multiple layers, genomic, epigenetic, proteomic, and microenvironmental, requiring integrative strategies that transcend single-marker approaches [16]. Moving forward, a multi-level framework that incorporates multi-omics profiling, functional readouts, immune and microenvironmental signatures, and liquid biopsy monitoring will be essential for accurate subtype classification and therapeutic tailoring. As precision oncology continues to expand, the combination of such cellular therapies with multi-omics profiling, digital pathology, and longitudinal liquid biopsy monitoring may further enhance the ability to anticipate resistance and tailor treatments at an individual level. Ultimately, advancing the study of emerging subtypes will refine prognostic accuracy while simultaneously unlocking novel therapeutic avenues, thereby improving survival and quality of life for patients facing this heterogeneous disease.

## Figures and Tables

**Figure 1 ijms-26-11599-f001:**
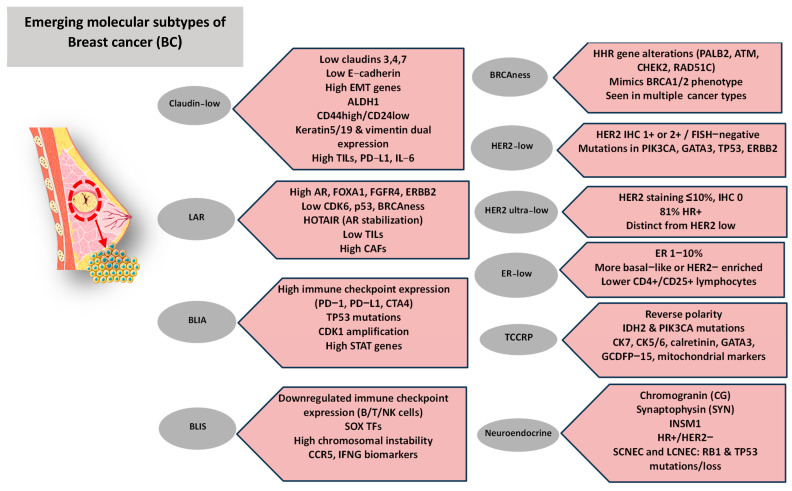
Emerging molecular subtypes of breast cancer (BC) and their defining features, highlighting distinct genomic alterations, signaling pathways, immune profiles, and phenotypic markers that refine current classifications and may guide therapeutic strategies.

**Figure 2 ijms-26-11599-f002:**
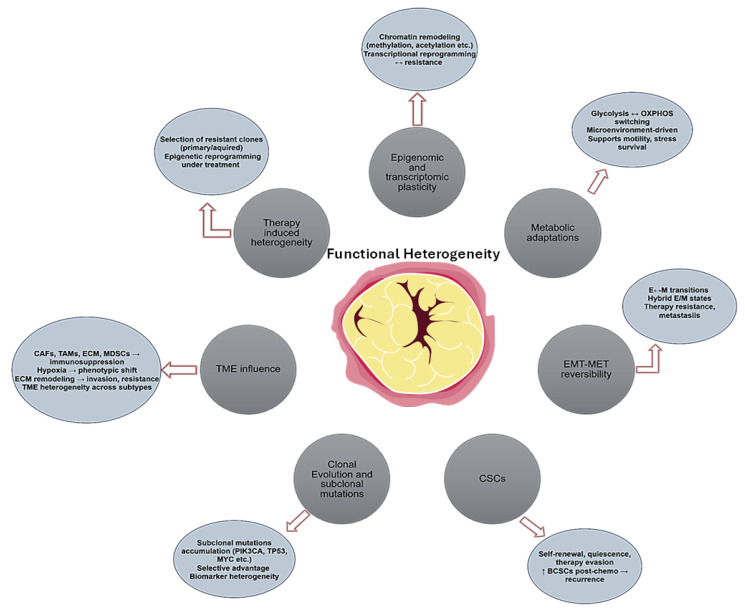
Functional heterogeneity in breast cancer (BC) and its prognostic impact. The illustration depicts cellular diversity, phenotypic plasticity, and tumor microenvironment (ΤΜΕ) interactions, emphasizing how these processes shape clinical outcomes and influence therapeutic response.

**Figure 4 ijms-26-11599-f004:**
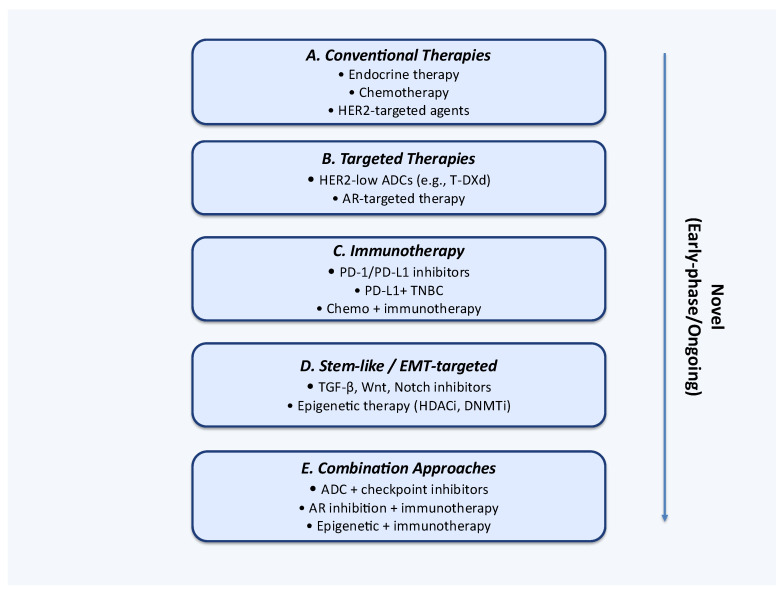
“Overview of current and emerging therapeutic strategies in Breast cancer”. The diagram summarizes key treatment categories, including conventional therapies, targeted agents, immunotherapy, stem-like/EMT-targeted approaches, and combination regimens. The vertical axis illustrates the progression from established treatments toward novel and early-phase strategies.

**Table 1 ijms-26-11599-t001:** Classical Breast Cancer (BC) Subtypes: Molecular Characteristics, Prognostic Relevance, and Therapeutic Implications.

Subtype (IHC/Gene Expression)	Typical Markers	Prognosis	Therapeutic Guidance	Limitations/Challenges	Detection Method/Sensitivity	Detection Method/Sensitivity
Luminal A	ER+/PR+, HER2−, Ki-67 low	Favorable	Endocrine therapy	Borderline Ki-67 may overlap with Luminal B; intra-subtype proliferation variability	IHC (≥1% positive nuclei)	Sensitivity ~90%; FP < 5%; FN ~10%
Luminal B	ER+/PR+, HER2+/−, Ki-67 high	Intermediate	Endocrine therapy ± chemotherapy	ER+/Ki-67-high tumors may behave aggressively; variable chemotherapy response	IHC + FISH (HER2 confirmation)	Sensitivity ~85%; FN 5–10%
HER2-enriched	HER2+, ER−/PR−	Poor without targeted therapy	HER2-targeted therapy	HER2-low tumors not captured; spectrum of HER2 expression ignored	IHC/FISH (score ≥3+ or gene amplification)	Sensitivity 95%; FP 3–5%
Basal-like/TNBC	ER−/PR−/HER2−, basal markers	Poor	Chemotherapy	Basal-like non-TNBC tumors overlooked; heterogeneity in chemosensitivity	IHC panel	Sensitivity 80–90%; variable inter-lab results

## Data Availability

No new data were created or analyzed in this study. Data sharing is not applicable to this article.

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
