# Peer review of "Emerging Breast Cancer Subpopulations: Functional Heterogeneity Beyond the Classical Subtypes"

_ijms, 2025, doi:10.3390/ijms262311599_

Round 1
Reviewer 1 Report
Comments and Suggestions for Authors
Emerging Breast Cancer Subpopulations: Functional Heterogeneity Beyond the Classical Subtypes
The review aims to provide a comprehensive overview of the emerging breast cancer subtypes including HER2-low, claudin-low, BRCA-deficient, luminal androgen receptor, and basal-like immune variants. The central thesis of the review is that the classical receptor-based classification of breast cancer subtypes fails to address the functional heterogeneity and that recognizing these emerging subtypes is essential for patient stratification and precision oncology. Overall, the authors have covered several aspects of the breast cancer biology with decent coverage of new publications (from last 5 years). From a reader’s perspective, the manuscript is a bit too long and repetitive with a long list of references. I think the authors should try to shorten the manuscript focusing on recent literacture.
Few major comments:
- References: Some of the references are wrong, please go through references and cite appropriately. Here is an example of a wrong reference ( Wai Fong Chua. Matters of concern and engaged research. Accounting & Finance, 2022, 62, 4615–4627). Similarly, please also check References 35, 38, 40,44, 45, 46, and 47. Bibliography list also has many mistakes like 68 doesn’t have author’s name but ends with pubmed (68. Unravelling triple-negative breast cancer molecular heterogeneity using an integrative multiomic analysis - PubMed). Similarly, also check 61-207. In line 523, 140 is written twice.
- Table 1 is incorrectly labelled as “Ongoing Clinical Trials for emerging BC Subpopulations” The trials listed don’t seem to be ongoing and they are not emerging BC subpopulations but classical BC subtypes.
- The introduction section and Classical BC subtypes and limitations have a lot of overlap. Merging them into a single section will increase the readability. In addition, the authors often list facts about subtypes but fail to provide a critical interpretation.
- The abbreviation list is helpful but not all the abbreviations are listed there, please update the list.
Author Response
Dear Reviewer 1,
We sincerely thank you for your thorough evaluation of our manuscript and for the constructive feedback provided. We appreciate your recognition of the relevance of the emerging breast cancer subtypes discussed, as well as your positive comments regarding the coverage of recent literature.
We also acknowledge your concerns regarding the length, repetition, and extensive referencing. In response, we have carefully revised the manuscript to improve clarity and flow, streamlined repetitive sections, and shortened content where appropriate. We have also refined the reference list to maintain focus on the most recent and impactful studies.
We believe these revisions have enhanced the readability and overall quality of the manuscript, and we thank you once again for your helpful suggestions.

Reviewer 2 Report
Comments and Suggestions for Authors
The manuscript with ID: ijms-3930199 titled “Emerging Breast Cancer Subpopulations: Functional Heterogeneity Beyond the Classical Subtypes” is a Review work where the authors discussed about the more recent outcomes found in the field of breast cancer and how the onset and progression of this disease can be influenced by the tumoral microenvironment. Furthermore, biomarkers to tailor the next-generation of smart customized therapies are also indentified. This is a topic of growing interest and the manuscript is generally well-written.
However, it exists some points that need to be addressed (please, see them below detailed point-by-point) to improve the scientific quality of the submitted manuscript paper before this article will be consider for its publication in the International Journal of Molecular Sciences.
1) Introduction. “Breast cancer (BC) remains the most frequently malignancy in women (…) approximately 7.8 million women were diagnosed within the last fiver years (…) 685,000 deaths recorded in 2020 alone” (lines 31-35). Could the authors provide quantitative data insights according to the worldwide disability-adjusted life years (DALYs) related to breast cancer disease? This will significantly aid the potential readers to better understand the significance of this Review work.
2) Then, some state-of-the-art concerning the breast cancer in men should be also stated in the Introduction section.
3) Classical BC Subtypes and Their Limitations. Table 1 (line 158). The detection limits and accuracy (in contrast to false positive/negative tests) should be also furnished for all the shown breast cancer biomarkers.
4) Emerging Breast Cancer Subpopulations. Figure 1 (line 375). The figure content is slightly blurry. The lettering size should be enlarged.
5) Functional Heterogeneity Within Breast Cancer Subtypes (lines 382-492). Here, the content provided in this subsection is accurate. No actions are requested from the authors.
6) Prognostic and Predictive Biomarkers in Aggressive Populations (lines 493-663). In this subsection, it should be also remarkable to briefly discuss about the emerging nanoscale imaging techniques [1] and how they can discriminate the nanomechanics among healthy and malignant breast cancer cells [2]. Furthermore, recent research devoted to design and develop machine learning methods for breast cancer prognosis should be also shown in this subsection.
[1] https://doi.org/10.1002/smsc.202500351
[2] https://doi.org/10.1016/j.actbio.2023.01.011
7) Therapeutical Implications and Ongoing Clinical Trials (lines 664-725). A schematic representation will also benefit the potential readers to identify and differentiate among the existing clinical therapies against breast cancer disease.
8) “7. Challenges and Future Directions” (lines 726-757) and “9. Conclusions” (lines 801-816). These sections perfectly remark the most relevant outcomes found by the authors in this field and the promising future prospectives. It may be also opportune to highlight the potential future action lines to pursue the topic covered in this work (e.g. CAR-T therapies, …).
Author Response
Dear Reviewer 2,
We thank you sincerely for your constructive evaluation of our manuscript and for highlighting the relevance and clarity of our work. We appreciate your positive remarks regarding the importance of the topic and the overall quality of the writing.
We also acknowledge the points you have outlined for improvement. We have addressed each of your comments in detail, and all corresponding modifications are clearly marked within the revised text for your convenience. These revisions include clarifications of key concepts, refinement of sections on tumor–microenvironment interactions, and an enhanced discussion of biomarkers and next-generation therapeutic strategies.
We believe that these changes have strengthened the scientific rigor and readability of the manuscript. We are grateful for your helpful feedback and the opportunity to improve our work.

Round 2
Reviewer 2 Report
Comments and Suggestions for Authors
The manuscript can be accepted